# Assessment of the Role of Snowmelt in a Flood Event in a Gauged Catchment

**Jesús Mateo-Lázaro** [1,*] , **Jorge Castillo-Mateo** [2] , **José Ángel Sánchez-Navarro** [1] ,
**Víctor Fuertes-Rodríguez** [3] , **Alejandro García-Gil** [4] and **Vanesa Edo-Romero** [1]

[1] Department of Earth Sciences, University of Zaragoza, Pedro Cerbuna, 12, 50009 Zaragoza, Spain;
joseange@unizar.es (J.Á.S.-N.); vanesa_edo@hotmail.com (V.E.-R.)

[2] Department of Mathematics, University of Zaragoza, Pedro Cerbuna, 12, 50009 Zaragoza, Spain;
720193@unizar.es

[3] Department of Geography and Territorial Planning, University of Zaragoza, Pedro Cerbuna, 12,
50009 Zaragoza, Spain; 696825@unizar.es

[4] Geological Survey of Spain (IGME), Manuel Lasala 44, 9° B, 50006 Zaragoza, Spain; a.garcia@igme.es

* Correspondence: jesmateo@unizar.es; +34-(9)-690-609948

**Abstract:** An actual event that happened in the Roncal valley (Spain) is investigated and the results are compared between models with and without snowmelt. A distributed rainfall model is generated with the specific data recorded by the rain gauges of the catchment during the episode. To describe the process of water routing in the hydrological cycle of the basin, a model is used based on combinations of parallel linear reservoirs (PLR model), distribution by the basin, and tip-out into its drainage network configured using a digital terrain model (DTM). This PLR model allows simulation of the different actual reservoirs of the basin, including the snow and the contribution due to its melting which, in the model, depends on the temperature. The PLR model also allows for a water budget of the episode where, in addition to the effective rainfall contribution, the water that comes from the thaw is taken into account. The PLR model also allows determination of the amount of water that exists in the basin before and after the episode, data of great interest. When comparing the simulations with and without taking into account the thawing process, it is evident that the intervention of the snow reservoir has been decisive in causing a flood to occur.

**Keywords:** availability of water resources; snowmelt; floods; hydrological processes; PLR models; SHEE software

## 1. Introduction

There are numerous works carried out in recent years in different lines of flood research, but there are few that include melting processes as a direct or added cause to the flood phenomenon. The process of snowmelt is considered in this paper within a flood model with hydro-meteorological methods or models of rainfall-runoff. We have introduced the melting process as an additional contribution that flows through the hydrological system of the basin.

Hydrological processes related to snow (snowfall, snowpack, snowmelt) occur in large regions of the Earth. Snowmelt is one of the processes intervening in the hydrological cycle and interacting with many other processes. This article focuses on the phenomenon of melt-induced flooding and the changes it causes in the runoff regime, but this issue and its consequences have already been widely studied using a multitude of approaches [1,2].

The frequency and the variability of the discharge in rivers and their relationship with runoff coming from snowmelt is a recurrent topic even in recent studies [3–6]. Sensitivity studies of the

variables that intervene in the melting processes are another key issue, for example, Kult et al. [7] evaluates the sensitivity of a snowmelt runoff model with temperature input data, in a region with complex temperature elevation gradients.

Another phenomenon to be highlighted, which is also caused by the melting process, is the time lag in the water balance. This lag takes place between precipitation in the form of snow and the runoff, which is another of the variables we quantify in this article. Recently, some authors have shed some light on this topic [8,9]. Other authors have developed rain–snow separation methods within time series and their lag with runoff [10–12]. Pistocchi et al. [13] uses data from the temporal estimate of the snow cover thickness to study the time lag. DeBeer and Pomeroy [14] models the relationships between snow cover depletion and the influence of snow accumulation heterogeneity and melting energy. Xie et al. [15] proposes calibration methods for the time lag of the snow-runoff. Banasik and Hejduk [16,17] investigates an small agricultural basin showing significantly higher values of lag time for snowmelt events than for rainfall events. In that study the relationship between lag time of runoff and sediment output is also presented.

In addition to the methods proposed in Meng et al. [18], there are recent publications that use models developed to implement the snow-runoff process. The well-known SWAT model has also been widely used in recent publications [1,19–33]. Xie et al. [15] develops a progressive segmented optimization algorithm to calibrate the temporal variation parameters of the snowmelt-runoff model. DeBeer and Pomeroy [14] conducts simulations of snowfall-runoff in various environments of mountain basins. The most frequent parameters used in the models are analyzed by Martinec and Rango [34], concluding with the development of hydrological basin models accounting for the runoff due to the snow thawing and showing that substantial progress has been made by using models with different sophistication levels.

Other authors focus their research on the forecasting of melting processes and their consequences. Corripio and López-Moreno [35] performs prediction studies about thawing processes happening in the Spanish Pyrenean chain, which is the research location area of our article. In other cases, the studies focus on urbanized areas such as that of Berezowski and Chybicki [36], where the effect of snow melting on discharge forecasts is highlighted by carrying out a study in an urbanized basin clearly influenced by snow processes increasing the runoff.

We earlier referred to the relationship between the melting process and other processes. Erosion and sediment transport are processes that have a relationship with snow melting. In many cases, water melted from snow causes a greater erosion of the soil than rainwater [37] so it is appropriate to investigate the differences in soil infiltration capacity during periods of rain or snow melting. Soil erosion in agricultural areas during winter and spring is a problem in many countries, such as Norway [38], the USA [39], Belgium [40], the United Kingdom [41], Germany [42], and Russia [43]. In these areas, soil erosion during winter and spring depletes the nutrient-rich top layer and contributes elements (phosphorus, nitrogen) to freshwater bodies [44].

The period of droughts is another process that may be related to the snow melting processes, in this case due to an absence or prolonged decrease in the contribution of snow [45].

Energy and thawing processes are also related to each other. Stock et al. [46] shows how changes in the composition or structure of the soil in large areas influence the energy input to the soil and, therefore, the rate of thawing. DeBeer and Pomeroy [14] demonstrates the influence of the spatial distribution of the snow cover thickness and heterogeneity on the distribution of fusion energy in extreme runoff events.

It can be said that the snow and thawing processes play an important role in the hydrological cycle, both by themselves and by their direct relationships with other processes, as well as by their regulatory effect on water flow and volume of the water reserve.

To carry out this work, we start with a flow circulation model using a methodology based on a combination of parallel linear reservoirs (PLR model) [47,48]. In hydrological science, reservoir models have been widely used to represent different characteristics of river basins. There are some

publications [49–51] that synthesize several works in this area of research. Currently, reservoir models are very widespread and are based on the concept that a watershed is like a set of interconnected deposits (rain, snow, aquifers, soil, biomass, etc.), each with different characteristics in terms of recharge, storage, and discharge [52]. The input hydrograph corresponds to the effective precipitation, also called effective rainfall, rainfall excess, or recharge. This is why this model has to be combined with another one performing the transformation of the gross precipitation into effective precipitation. It is very usual, and it is necessary in our case, to use the curve number method of the SCS (SCS–CN model) to carry out this transformation. Other authors have made this same decision and have used the SCS–CN model for the same purpose [53].

The models of reservoirs are traditionally used in hydrology to represent different characteristics of the basins, and Dewandel et al. [51] presents a synthesis of different works in this line. Recently, a line of research has been established with models of reservoirs distributed according to geomorphological trajectories of flow within the basin. For example, in Boyd [54], a conceptual model is developed based on the geomorphological properties of watersheds. Unitary hydrograph models applying linear reservoirs distributed in cascade according to the geomorphology of the basin are developed in López et al. [55].

This article investigates the cause of an actual flood event that occurred in the Roncal valley in January 2009, where everything seems to indicate that fusion processes in the basin's snow reservoirs contributed decisively to the activation of the flood. The work is carried out following the methodology developed in Mateo-Lázaro et al. [47]. The computer program SHEE (Simulation of Hydrological Extreme Events, as found in http://www.unizar.es/hidrologia) is used to do so, allowing representation of the basin through a distribution of reservoirs in parallel (PLR model) and taking into account the contributions due to the snow melt.

With regard to the investigation of the actual episode, the following scheme is followed: (1) Creation of a distributed model of total rainfall. (2) Creation of a water circulation model for the basin. The parameters of a reservoir model ($Q_{oi}$ and $\alpha_i$) are determined from the recession curves of the actual hydrograph. (3) Creation of a precipitation model and generation of the hyetographs of total precipitation using the curve number model, another effective precipitation model. (4) Establishment of the water balance. Based on the results of the simulation, a water balance is calculated that determines the origin of the water causing the flood.

## 2. Methods

The PLR models are part of the SHEE (Simulation of Hydrological Extreme Events) computer program, carried out in the Department of Earth Sciences of the University of Zaragoza for its application in hydrological events. In essence, the program is an adaptation of traditional hydrological models to new technologies and data sources and reproduces, through the use of models, the different intervening processes in water flow. In general, these processes are the geometric model of the basin; the rain model that distributes it in space and time; the rain-runoff transformation model that aims to differentiate (or separate) between losses and runoff (losses are also called abstractions, retentions, and runoff deficit); and the flow circulation models (also called transit, propagation, routing, runoff, and discharge). Then, to this general scheme, a new model is added that estimates the contribution of water coming from snow reservoirs.

Figure 1 shows the interface of the computer program which has been used in recent years in numerous professional works on risks, spatial planning, and civil and architectural works, as well as in scientific publications [47,48,56–63]. The SHEE software has numerous applications for either DEM management or simulation of hydrological processes [64]. Obtaining new cartographic coverage from the combination of DEM and simulated processes is also possible. DEM management is achieved using the GDAL (Geospatial Data Abstraction Library). The program can combine coverage from different coordinate systems thanks to the use of the PROJ4 library from the USGS. Downloading information from the WMS remote server is also possible. With regard to the DEM characteristics, the SHEE

program can manage any format, size, accuracy, and reference system. For example, Global DEM has been used as SRTM30 (with file size 3.6 GB and grid size 30″, ≈900 m), MDT5 of Spanish territory (120 GB and 5 m), and LIDAR. The use of PLR (parallel linear reservoir) models as hydrological models integrated within the sequential processing algorithm of the catchment is a special case of hydrological application where every cell of the DEM is considered as a reservoir combination in parallel [65].

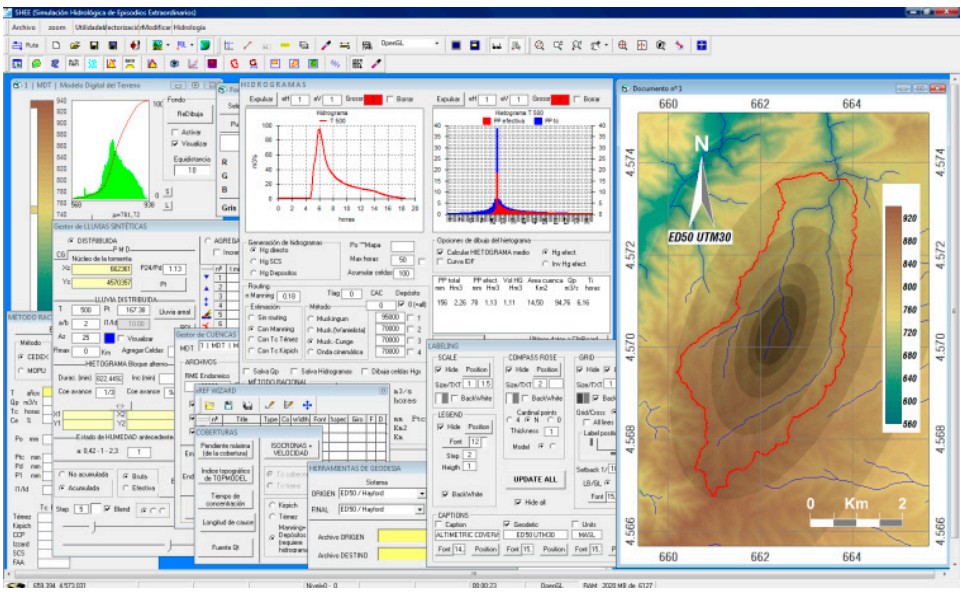

**Figure 1.** The Simulation of Hydrological Extreme Events (SHEE) program interface. This program uses digital terrain models and hydrological models, and actual or simulated rain, soil moisture, and circulation through the drainage network.

The future developments of the SHEE program are very promising. Recently, we have incorporated a module for the development and visualization of geological structures in three dimensions. In the near future, it will allow development of combined hydrogeological models with complex subsurface structures. This implementation has resulted in the publication of Mateo-Lázaro [58]. Therein, we present an application that visualizes three-dimensional geological structures with digital terrain models. The three-dimensional structures are displayed as their intersection with two-dimensional surfaces that may be defined analytically (e.g., sections) or using grid meshes in the case of irregular surfaces such as the digital terrain models. Additionally, the process of generating new textures can be performed by a graphics processing unit (GPU), thereby making real-time processing very effective and providing the possibility of displaying the simulation of geological structures in motion. Regarding the graphics processing units, and since the DEM is becoming denser, we are currently completing the development of hydrological models with this technique through the sequential processing algorithm, whose main advantage is the shortening of the computation time, which can be reduced up to 100 times. Due to parallel processing, it is necessary to reprogram the sequential algorithms for computing drainage networks.

## 2.1. PLR Models: Characterization of a Single Linear Reservoir

The hydrological relationships of a linear reservoir are governed by two equations, the flow or storage equation (Equation (1)) and the continuity equation or water balance equation (Equation (2)).

$$Q = \alpha \cdot S \tag{1}$$

$$R = Q + \frac{dS}{dt} \tag{2}$$

where Q is the discharge (m$^3$/sec), S is the storage (m$^3$), $\alpha$ is the depletion coefficient (sec$^{-1}$), R is the recharge that enters the reservoir (for example, in the form of effective precipitation), and t is the time (sec). From a topology perspective, the depletion coefficient ($\alpha$) is the slope value relating the storage with the discharge (Figure 2) and, hence, the name of the linear reservoir. The combination of Equations (1) and (2) results in the differential equation of runoff or discharge (Equation (3)).

$$Q_2 = Q_1 \cdot e^{-\alpha \Delta t} + R \cdot (1 - e^{-\alpha \Delta t}) \tag{3}$$

where $Q_1$ and $Q_2$ are the values of Q separated by an elementary fraction of time ($\Delta$t), during which the recharge can be considered constant.

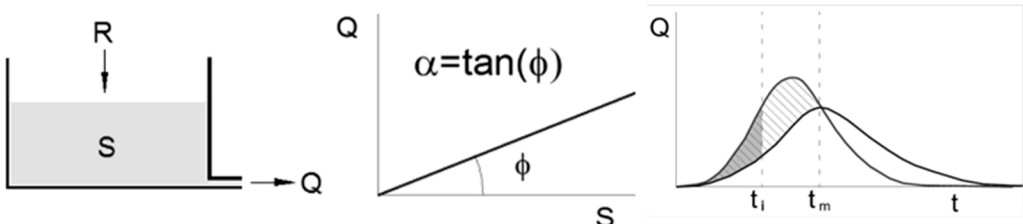

**Figure 2.** Water routing for a linear reservoir.

### 2.2. PLR Models: Combination of Linear Reservoirs in Parallel Sets

A basin (or a cell considering the areal division of the basin as cells of a grid) can be represented with a single reservoir or with a combination of reservoirs. The simplest model to represent a basin is to consider a single reservoir whose behavior is representative of the basin's response. Evidently, the model is not very descriptive and the results obtained show big deviations. In addition, a basin can be represented by a set of deposits, each one representative of certain characteristics of the basin. Furthermore, a basin can be divided into sub-basins or cells, each of which can be represented by a set of deposits. The possible combinations of reservoirs to build a model are unlimited, although few combinations give acceptable results. In this work, we have used a combination of two reservoirs in parallel for each cell of the digital terrain model, where water enters according to a distributed model of effective precipitation. The combination of both reservoirs pours water into the drainage network, whose flow is modeled with a classical method. This scheme is used in the SHEE program to meet the following assumptions:

(1) The calibration of the model is based on actual recession curves, i.e., based on actual responses from the basin.

(2) Each reservoir represents a set of characteristics for a specific basin, whose response differs markedly from the response of the other reservoirs.

(3) As a result of assumption (2), the total hydrograph can be decomposed into several partial hydrographs that can be mapped into parts of the hydrological system (direct runoff hydrograph, underground runoff hydrograph, etc.).

(4) The PLR model has the ability to ascertain the water stored in the reservoirs (i.e., in the basin) and to calculate a water balance for each episode.

(5) Finally, standard models (kinematic wave or Muskingum–Cunge) are compared to contrast the circulation in channels.

For a combination of n linear reservoirs in parallel, the result is the sum of the reservoirs as shown in Equations (4) and (5); and for each reservoir, when using Equations (1) and (2) without considering recharge, Equations (6) and (7) are obtained.

$$Q = \sum_{i=1}^{n} Q_i \tag{4}$$

$$S = \sum_{i=1}^{n} S_i \tag{5}$$

$$Q_i = Q_{0i} \cdot e^{-\alpha \Delta t} \tag{6}$$

$$S_i = \frac{Q_i}{\alpha_i} \tag{7}$$

Equation (8) is obtained by combining Equations (4) and (6). Equation (8) shows that, for an instant t, there is a mathematical relationship between the flows of each deposit.

$$\left( \frac{Q_1}{Q_{01}} \right)^{\frac{1}{\alpha_1}} = \ldots = \left( \frac{Q_i}{Q_{0i}} \right)^{\frac{1}{\alpha_i}} = \ldots = \left( \frac{Q_{nr}}{Q_{0nr}} \right)^{\frac{1}{\alpha_{nr}}} \tag{8}$$

The recession curve of a hydrograph represents data flow vs. time; $Q_i - t_i$ in the case of the system of Equation (8), whose resolution allows a set of $Q_{oi}$ parameters, $\alpha_i$, to be obtained. With these parameters, a PLR model can be established. Once the model is established, hydrographs can be generated using Equations (3), (4), and (6), according to the inputs ($R_i$) to the system. In addition, using Equations (5) and (7), the amount of water stored in each reservoir of the basin is obtained, thus allowing the determination of the water balance.

*2.3. PLR Models: Snowmelt*

Snow thawing implies a new contribution that needs to be added to the input along with the effective precipitation, and whose origin has to be looked for in a previous episode of precipitation in the form of snow. For a given process of snowmelt due to short events of intense rain, precipitation and temperature are considered the most influential parameters in the fusion process. A simple model will consider snow melting as linearly proportional to precipitation when occurring at temperatures above 0 °C. Thus, for a certain part of the hydrograph that coincides with temperatures above 0 °C, effective precipitation as an input to the surface runoff system can be replaced by total precipitation by adding snow melting. In this way, the runoff equation (Equation (3)) is modified by adding a new recharge term coming from the snow melting ($R'$), and by substituting the effective precipitation with total precipitation at R, as expressed in Equation (9).

$$Q_2 = Q_1 \cdot e^{-\alpha \Delta t} + (R + R') \cdot (1 - e^{-\alpha \Delta t}) \tag{9}$$

The main advantage of the PLR model is that, by allowing the water budget of an event to be considered, the amount of snowmelt recharge, $R'$, can be estimated to equilibrate the water balance.

## 3. Data Sources

*3.1. Characterization of the Basin*

The catchment studied is that of the Esca river (Roncal valley), located in the Pyrenees ridge, between Aragon and Navarra, two regions of northeast Spain. The basin is preserved in a natural state, without reservoirs or significant alterations. Figure 3 shows an altimetry representation of the basin, the location of the gauging stations (two), and the pluviometry stations considered (four). Table 1 gives some of the most significant hydrological characteristics for this watershed.

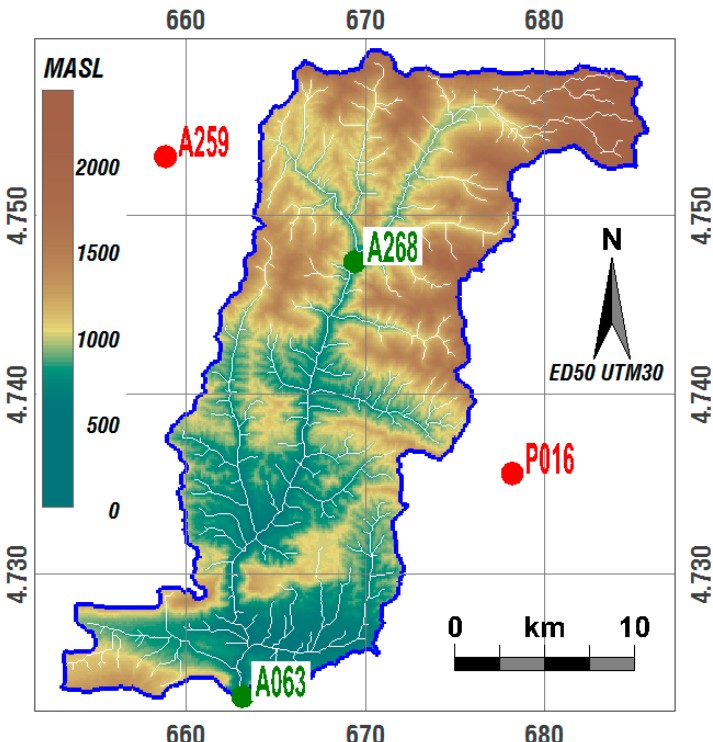

**Figure 3.** Altimetry representation of the Esca river basin with rain gauges (A063, A268, A259, and P016) and two stream gauges (A063, A268) that also have rain gauges.

**Table 1.** Hydrological characteristics of the Esca river basins in Sigues and Isaba.

| Characteristic | ud | Isaba (A268) | Sigues (A063) |
|---|---|---|---|
| Catchment area | km$^2$ | 189.06 | 506.26 |
| Length of the main channel | km | 24.01 | 51.70 |
| Average slope of the main channel | % | 6.44 | 3.45 |
| Average slope of the digital surface of the catchment | % | 36.85 | 34.88 |
| Average curve number (AMC II) | | 63.08 | 61.22 |

## 3.2. Melting Event Setting

The event investigated corresponds to the flood that occurred between January 18 and 28, 2009. Table 2 shows its main characteristics, such as duration, peak flow, and the number of intervals considered for the treatment of the actual rain.

**Table 2.** Characteristics of the actual event investigated.

| Date | | Rain Duration (h) | Time Interval (15 min) | Peak Flow(m$^3$/s) |
|---|---|---|---|---|
| Start Time | End Time | | | |
| 2:30 p.m. 18 January 2009 | 7:30 p.m. 28 January 2009 | 245.25 | 981 | 201 |

## 3.3. Rainfall Model

The rainfall data has been obtained from the Spanish Hydrological Information Warning System SAIH, network that records the value of the precipitation in 15-minute intervals. Figure 4 shows two isohyet maps of the event corresponding to the coverage generated from the accumulated precipitation in two of these intervals (the event requires the use of 981 layers). Spatial interpolation is performed by applying a distributed precipitation model based on the use of radial basis functions (RBF) [66–68].

From this precipitation coverage, the computer program can generate a different hyetograph for each point of the basin.

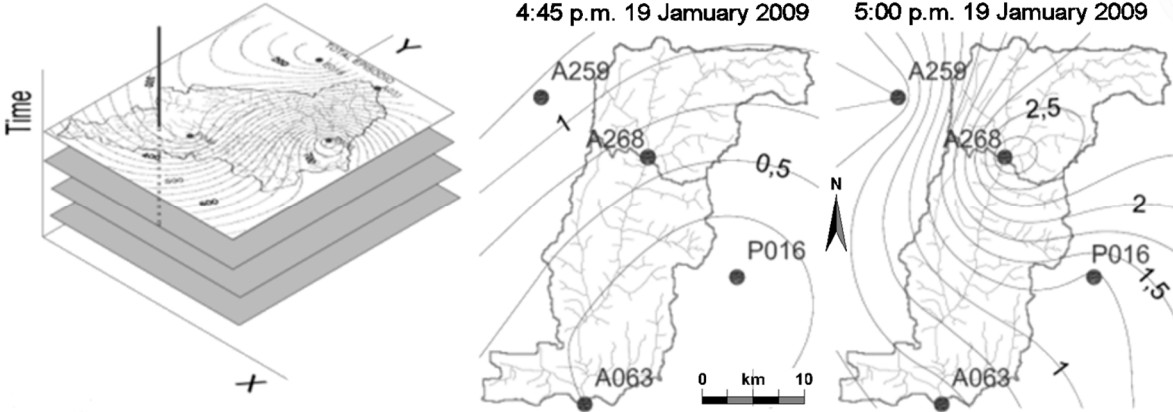

**Figure 4.** Isohyet maps of the event that occurred in January 2009 (equidistance 0.25 mm). A sequence of two coverage intervals of 15 min is presented. The whole event has 981 layers.

### 3.4. Reservoir Model

The reservoir model is established from the actual hydrographs of the episode by adjusting the parameters ($q_{0i}$ and $\alpha_i$). For this purpose, the least squares method is applied to the specific recession curve using Equation (10) (q = Q/catchment area).

$$q_i = \sum_{j=1}^{m} q_{0i} \cdot e^{-\alpha_j \cdot t_i} \tag{10}$$

where m is the number of reservoirs (two in our case), $q_j$ are the n known values of the flow of the recession curve at time $t_i$, and $q_{0j}$ and $\alpha j$ are the unknowns. Therefore, an overdetermined system of n equations is obtained, with 2m unknowns, where n >> 2m. Once stated and solved for the recession curve of the actual hydrograph, the results shown in Table 3 are obtained for a model of two linear reservoirs in parallel, which will be used within the SHEE program to perform simulations, either considering or dismissing the snowmelt process.

**Table 3.** Parameters of the parallel linear reservoirs for the event that occurred in the Esca river.

| Parameter | Reservoir | |
|---|---|---|
| | 1 | 2 |
| a → | $1.43 \times 10^{-6}$ | $1.67 \times 10^{-5}$ |
| $q_0$ → | $2.96 \times 10^{-7}$ | 1.00 |

## 4. Results and Discussion

### 4.1. Simulation without Snowmelt

The simulation of the event using the SHEE program and not considering snowmelt shows a notable mismatch with reality. To achieve an adequate water balance, modifying the standard conditions to very wet conditions by means of the curve number model is required, as seen in the ratio of total and effective rainfall hyetographs (Figure 5). It is evident that the effective rain obtained is unusually high. On the other hand, there is very little agreement between both hydrographs (Figure 6). In the actual hydrograph, a peak occurs at 125 h that does not occur in the simulation, while at 220 h, the simulated values exceed the actual ones.

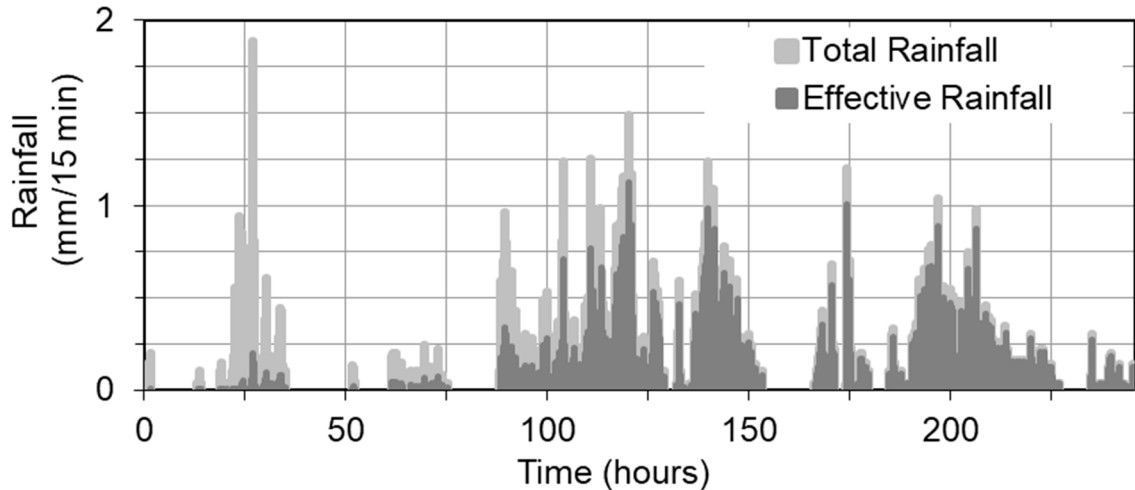

**Figure 5.** Average hyetographs in the Esca river basin in January 2009. Total precipitation was obtained directly by the interpolation method (light grey) and effective precipitation (dark gray), not considering snowmelt and with high curve numbers.

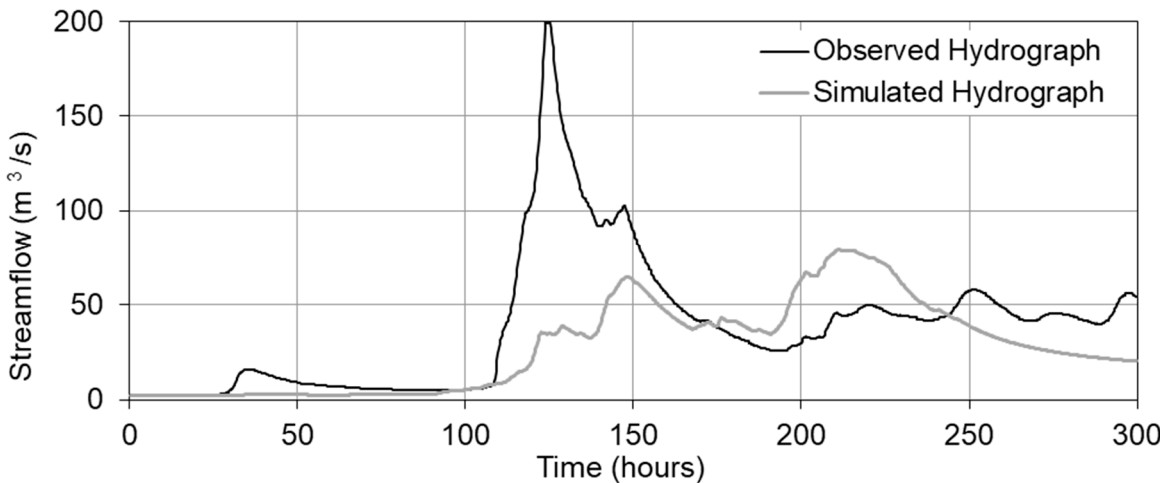

**Figure 6.** Hydrographs of the January 2009 event of the Esca river in Sigues (A063 stream gauge), actual and simulated, not considering snowmelt.

### 4.2. Simulation with Snowmelt

In the model considering snowmelt, the average altitude of each of the two basins was taken into account, as well as the discharge contributions into each stream gauge. It has been observed that there is no proportional relationship between the area of each basin and its discharge contribution. This fact was to be expected, because the melting is greater at lower altitudes in winter, a circumstance that tends to be reversed towards spring, as there is more snow in the high areas.

The ratio of the peak-flow with a rapid melting of snow is supported by the increase of temperatures, a fact that can be seen in Figure 7. The station of Arangoiti is located at 1350 m with a cold period between 45 and 90 h, while between 90 and 140 h the temperature exceeds 0 °C, reaching up to 8 °C.

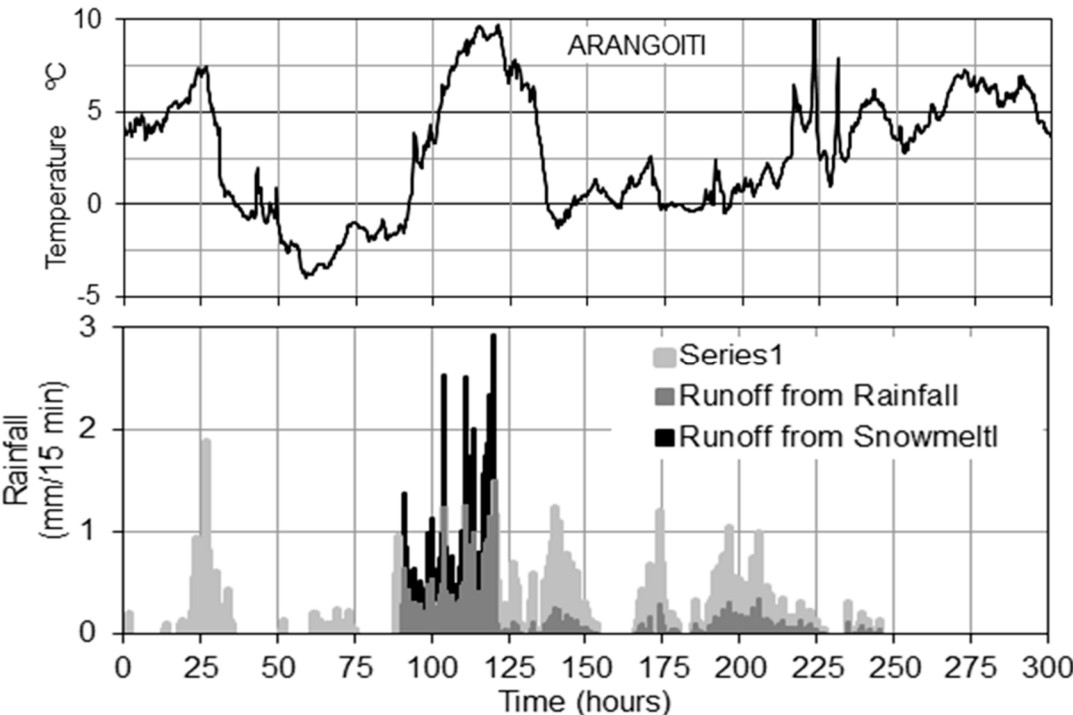

**Figure 7.** Temperature graph at the Arangoiti station during the episode of January 2009 and hyetographs of total and effective precipitation with snowmelt in the Esca river.

By using the model, the necessary recharge derived from snowmelt, which is needed in the interval between 80 and 120 h to fit the hydric balance of the episode in less humid hydrological conditions than in the case of the no snowmelt model, has been estimated. The optimal results are obtained with a melting value equivalent to 90% of the gross precipitation that occurred in that interval, that is, $R' = 0.9 \cdot R$. In this way, the hyetographs shown in Figures 7 and 8 are obtained, where previously dry hydrological conditions have been considered. Thus, in the simulation, a hydrograph very similar to the actual one is obtained.

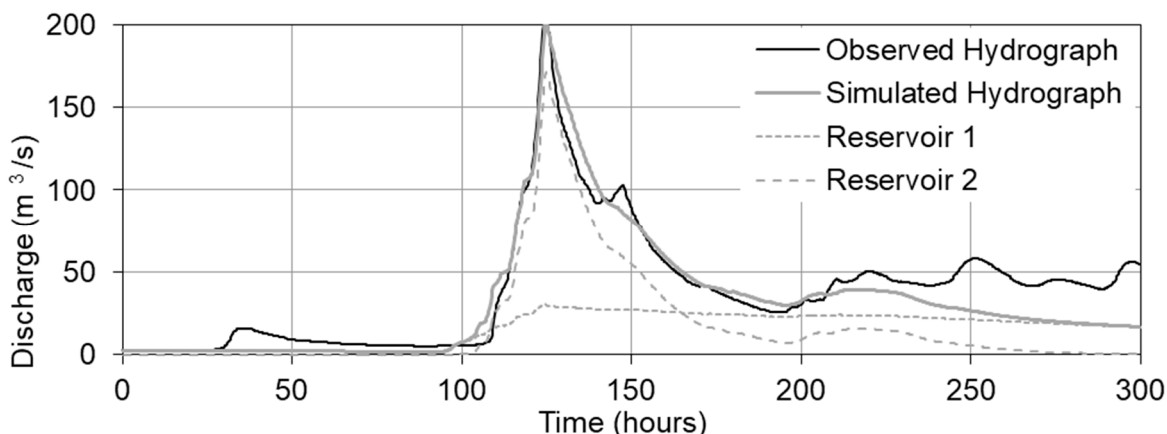

**Figure 8.** Simulated hydrograph considering snowmelt. Fragmented lines show the partial hydrographs for each reservoir.

Figure 9 shows the observed hydrograph obtained from the Isaba stream gauge (A268), which is located in the interior of the basin and can be compared with the Sigues stream gauge (A063) at the exit of the basin. Figure 10 shows the simulated hydrograph at the point corresponding to the Isaba station.

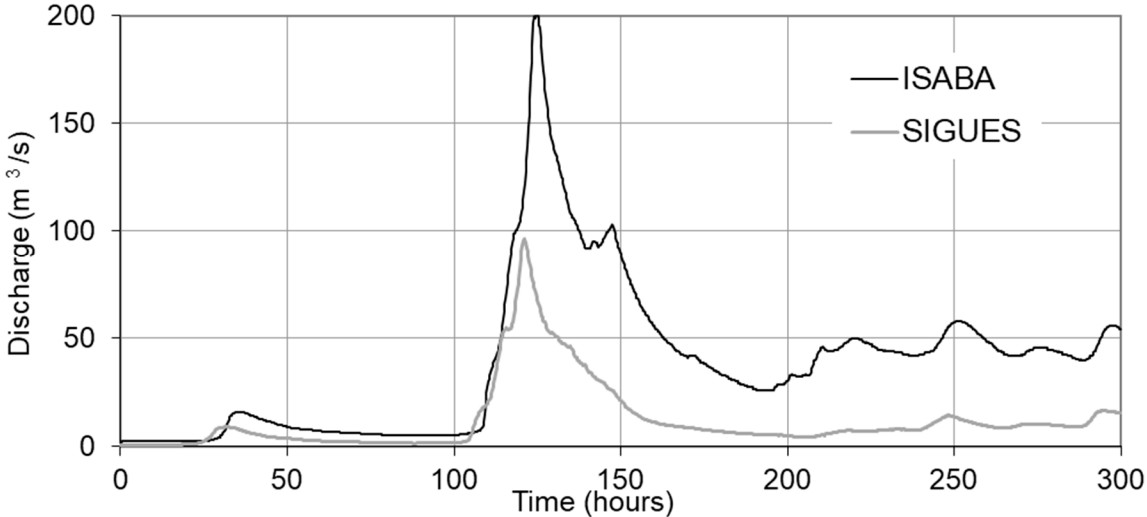

**Figure 9.** Actual hydrographs obtained from the Isaba (A268) and Sigues (A063) stations.

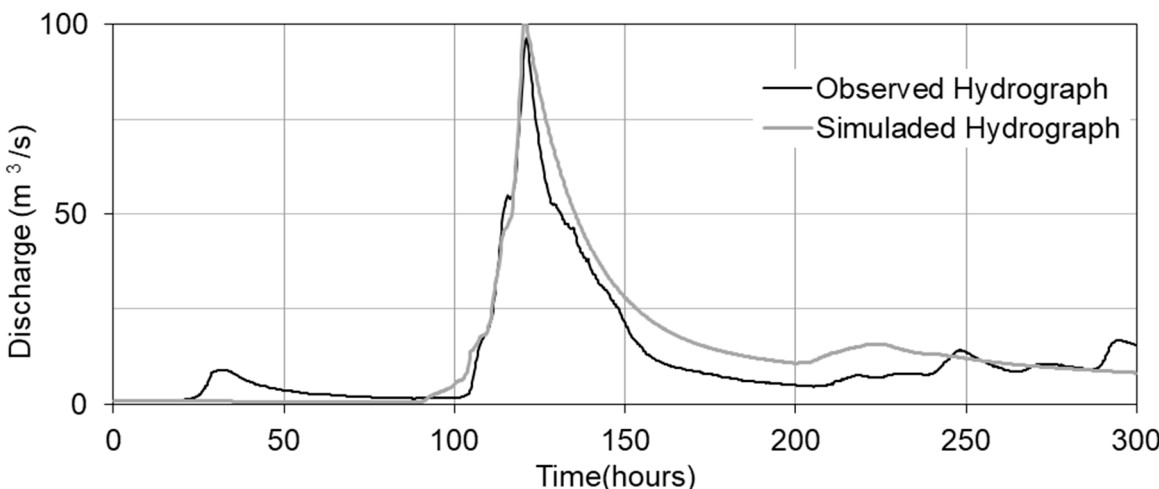

**Figure 10.** Hydrographs observed and simulated at the Isaba stream gauge (A268).

### 4.3. Water Budget

Table 4 contains the results of the water balance of the episode studied, considering the temporal interval between 0 and 200 h. The balance is carried out from the two gauging stations (Isaba and Sigues) and the results are given in total units (hm$^3$) for the entire basin corresponding to each gauging station. The results are also presented in specific units (mm), i.e., total units divided by the area of the corresponding basin. The last column represents a comparison between the results of the two gauging stations.

To elaborate on the water balance, we use the data and results of the hyetographs and the hydrographs used in the modeling, which can be summarized in three components:

(1) Total precipitation (Tp): it was obtained from the model of distributed precipitation based on functions (RBF) shown in Figure 4, adding all the 15-minute time intervals between 0 and 200 h. The results of this integration match the term 'total rainfall' from the average hyetographs of Figures 5 and 7.

(2) Snowmelt (Sm): it was determined in the distributed model calculated with the SHEE program, whose average representation matches the snowmelt fraction of the average hyetograph shown in Figure 7.

(3) Total runoff (Tr): it was also obtained from the distributed model, and the average value is represented in the hyetograph shown in Figure 7 as the sum of two terms: runoff from rainfall and runoff from snowmelt.

**Table 4.** Benchmarking of the water budget (within the range 0–200 h) in the stream gauges of Isaba and Sigues.

| Acronym | Tilte | Isaba hm$^3$ | Sigues hm$^3$ | Isaba mm | Sigues mm | Isaba/Sigues % |
|---|---|---|---|---|---|---|
| Tp | Total precipitation | 31.79 | 67.61 | 168 | 134 | 127% |
| Tr | Total runoff | 17.90 | 42.06 | 95 | 83 | 90% |
| Ep | Effective precipitation | 11.47 | 19.27 | 61 | 38 | 101% |
| Sm | Snowmelt | 6.43 | 22.79 | 34 | 45 | 76% |
| L | Losses | 20.32 | 48.33 | 107 | 95 | 148% |
| Hv * | Hydrograph volume | 9.80 | 25.86 | 52 | 51 | 80% |

\* Volume measured between 0 and 200 h. The water volume still has to leave the basin (in the Sigues stream gauge) $42.06 - 25.86 = 16.2$, calculated by adding the volume prior to the start of the event.

In short, the terms Tp, Tr, and Sm are obtained from data and direct results are obtained from the model. The terms Ep and L can be obtained from Equations (11) and (12), which are partial water balance equations:

(1) Effective precipitation (Ep): in Figure 7 it is described as runoff from rain.

$$Ep = Tr - Sm \tag{11}$$

(2) Losses (L): the difference between total precipitation and effective precipitation.

$$L = Tp - Ep \tag{12}$$

The volume of the hydrograph (Hv) is obtained by integration of the actual hydrograph within the 0–200 h range. The volume of water remaining within the basin at the 200 h coordinate is the difference between the total runoff (Tr) and the volume of the hydrograph (Hv), calculated by adding the volume present prior to the start of the episode. By examining Table 4 and analyzing the specific values (mm), the following statements are deduced:

(1) Based on the observed data: the Isaba basin shows a greater proportion of gross precipitation, although the output is similar to the Sigues basin (considering the output volume is equal to the volume observed in the hydrographs).

(2) Based on data from the simulation, at the Isaba basin the higher the average altitude of the basin, the lesser the thawing and the greater the losses. Part of those observations can be attributed to snow retention.

## 5. Conclusions

The hydrological modeling of watersheds by combining deposits is considered a classic method, but it is in recent years that it has become more widely spread with the development and improvement of computers. In this study, a new model was presented that combined linear deposits in parallel, representing the circulation of water through the reservoirs of the basin and giving excellent results in simulations of actual events. A quality that stands out about this model is its calibration, which is based on actual recession curves that are the response to the discharge of all the reservoirs of the basin. Furthermore, this model allows introduction of the discharge of water from the snow reserves, its distribution in the different deposits, and its routing through the hydrological system of the basin. Another additional advantage of the model is that it allows establishment of a water balance for an event and, thus, it estimates the recharge volume from the snowmelt.

The SHEE computer application allows configuration of models of hydrological processes, such as the spatial–temporal distribution of rainfall, the previous state of soil moisture, and the mode of water

routing through the basin. All the aforementioned factors have made it possible to carry out this investigation of an actual flood episode registered in the Esca river, where snow melting processes have had an outstanding influence.

If the snowmelt is not considered, the January 2009 event of the Esca river presented an extraordinarily unbalanced water budget. That is why a basic thawing model is introduced that is activated in periods of time where the temperature, taken from actual records, is higher, resulting in a congruous water balance and a better adjusted simulated hydrograph. Based on the simulation of this episode, the following observations are derived:

(1) An estimate of the melted snow volume (given as volume of water) is obtained, being 6.43 hm$^3$ at the Isaba stream gauge and 22.79 hm$^3$ at the Sigues stream gauge. The effective precipitation volumes are 11.47 and 19.27 hm$^3$, respectively. That way, the snowmelt represents 36% and 54% of the total runoff (effective precipitation plus snowmelt) respectively.

(2) With regard to the catchment area, the Isaba stream gauge produces greater total precipitation but lower runoff volume.

(3) The runoff due to thawing is notably higher at the Sigues station (45 mm compared to 34 mm at Isaba). This can be explained due to the snow melting occurring in the lower part of the catchment.

The importance snowmelt possesses in the genesis of floods is highlighted in this paper. To do this, another simulation is carried out without taking into account the thawing process. In that case, without snowmelt a peak flood of only 80 m$^3$/s is obtained, with a total runoff volume of 19.27 hm$^3$, compared to the 201 m$^3$/s and the 42.06 hm$^3$ obtained when considering the snowmelt.

**Author Contributions:** Conceptualization, J.M.-L. and J.C.-M.; methodology, J.M.-L. and J.Á.S.-N.; software, J.M.-L.; validation, J.M.-L., J.C.-M. and A.G.-G.; formal analysis, J.Á.S.-N., V.F.-R. and V.E.-R.; investigation, J.M.-L. and J.C.-M.; resources, J.M.-L., J.Á.S.-N. and A.G-G; data curation, J.M.-L., J.C.-M. and V.F.-R.; writing—original draft preparation, J.M.-L. and J.C.-M.; writing—review and editing, J.C.-M., A.G.-G., V.F.-R. and V.E.-R.; visualization, J.M.-L.; supervision, J.M.-L.; project administration, J.M.-L. and J.Á.S.-N; funding acquisition, J.M.-L. and J.Á.S.-N.

**Funding:** This work has been partially subsidized by the Research Group "Analysis of Continental Sedimentary Basins" of the Government of Aragon and FEDER Funds to which gratitude is extended.

**Acknowledgments:** We deeply appreciate the time spent by the reviewers and editors, who will undoubtedly contribute to the improvement of our manuscript.

**Conflicts of Interest:** The authors declare no conflict of interest.

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
