# Peer review of "Assessment of the Role of Snowmelt in a Flood Event in a Gauged Catchment"

_water, doi:10.3390/w11030506_

Round 1
Reviewer 1 Report
The question is not well defined. The methods and software are not described with sufficient details to allow readers to understand the results. It could be interesting to the readers, however it requires rearranging and rewriting.
Author Response
Dear Reviewers and Editors,
We would like to thank three anonymous Reviewers for having handled and reviewed our manuscript. We really appreciate your comments to improve this manuscript. We would like to show you what we have revised following your comments. In this letter, in blue you can read our reply to your kind comments.
In the file “water-427845_marked_manuscript”, the corrections are in red.
In the file "water-427845_Clean_manuscript_Sami revised" in blue, the changes of the language revision are marked.
The final version with all the changes is in the file "water-427845_Clean_manuscript"
Reviewer 1:
The question is not well defined. The methods and software are not described with sufficient details to allow readers to understand the results. It could be interesting to the readers, however it requires rearranging and rewriting.
Thanks for this suggestion. We have made a substantial reform of the paper, especially in the Introduction, Methods, Data Source, Results and Discussion sections and in Bibliographic References.
We have made an important structural and background reform in the introduction. The introductory framework and its references are carried out sequentially ordered by theme. The changes are the following:
“Hydrological processes related to snow (snowfall, snowpack, snowmelt) occur in large regions of the Earth. The snowmelt is one of the processes that intervene in the hydrological cycle interacting with many other processes. This article focuses on the phenomenon of melt-induced flooding and the changes it causes in the runoff regime, but this issue and its consequences has already been widely studied from a multitude of approaches [1, 2].
The tendency and the variability of the discharge in the rivers and their relation with the runoff coming from the snowmelt is a recurrent topic even in recent studies [3-6]. Sensitivity studies of the variables that intervene in the melting processes is another key issue, for example Reference [7] evaluates the sensitivity of a snowmelt runoff model with temperature input data, in a region with complex temperature elevation gradients.
Another phenomenon to be highlighted caused by the melting process is the time lag in the water balance, a lag that takes place between precipitation in the form of snow and runoff, which is another of the topics we quantify in this article. Recently other authors have contributed light on this topic [8, 9]. Other authors have developed rain-snow separation methods within time series and their lag with runoff [10-12]. Reference [13] uses data from the temporary estimate of the thickness of the snow cover to study the time lag. Reference [14] models the relationships of the depletion of the snow cover with the influence of the heterogeneity of the accumulation of snow and with the melting energy. Reference [15] proposes calibration methods for the time lag snow-runoff.
In addition to the methods proposed in [23] there are recent publications that use models developed to implement the snow-runoff process. The well-known SWAT model is widely used also in recent publications [1, 16-31]. Reference [15] develops a progressive segmented optimization algorithm to calibrate the temporal variation parameters of the snowfall-runoff model. Reference [14] conducts simulations of snowfall-runoff in various environments of mountain basins. The most frequent parameters used in the models are analysed by reference [32] which concludes with the development of hydrological basin models with runoff from the thaw and shows that substantial progress has been made with models of different sophistication.
Other authors focus their research on the forecast of melting processes and their consequences. Reference [33] performs prediction studies with thaw in the Spanish Pyrenean chain, research location area of our article. In other cases, the studies focus on urbanized areas such as Reference [34] where the effect of snow melting on discharge forecasts is highlighted, carrying out a study in an urbanized basin with clear influence of snow processes on the increase of runoff.
Above we have referred to the relationship of the melting process with other processes. Erosion and sediment transport are processes that maintain a relationship. In many cases water melted by snow causes greater erosion of the soil than rainwater [35] so it is appropriate to investigate the differences in soil infiltration capacity during periods of rain or melting. Soil erosion in agricultural areas during winter and spring is a problem in many countries of the world such as Norway [36], USA [37], Belgium [38], United Kingdom [39], Germany [40] and Russia [41]. In these areas, soil erosion during winter and spring depletes the nutrient-rich top layer and contributes elements (phosphorus, nitrogen) to freshwater bodies [42].
The period of droughts is another process that may be related to the melting processes, in this case due to an absence or prolonged decrease in the contribution of snow [43].
Energy and thaw are also processes related to each other. Reference [44] shows how changes in the composition or structure of the soil in large areas influence the energy input of the soil and therefore the rate of thawing. Reference [14] demonstrates the influence of spatial distribution of snow cover thickness and heterogeneity in the distribution of fusion energy in extreme runoff production.
It can be said that the snow and thaw processes play an important role in the hydrological cycle, both in themselves and by the direct relationship with other processes, as well as by their regulatory role of water flow and volume of water reserve.”
Section 4.3 has been rewritten, taking special care in the units and in the origin of the data used in the water balance:
“Table 4 contains the results of the water balance of the episode studied considering the temporal section between 0 and 200 hours. The balance is carried out in the two gauging stations (Isaba and Sigues), and the results are given in total units (hm3) for the entire basin corresponding to each gauging station. They are also given in specific units (mm), that is, total units dividing by the area of the corresponding basin. The last column represents a comparison between the results of the two gauging stations.
To elaborate the water balance, we start from the data and results of hyetographs and hydrographs that we have been using in modelling and that are summarized in three components:
(1) Total precipitation (Tp): It has been obtained from the model of distributed precipitation based on functions (RBF) shown in Figure 4, adding all the time intervals every 15 minutes from 0 to 200 hours. The results of this integration coincide with the term Total Rainfall of the average hyetograph of Figures 5 and 7.
(2) Snowmelt (Sm): it is determined in the distributed model calculated with the SHEE program, whose average representation coincides with the snowmelt fraction of the average hyetograph of Figure 7.
(3) Total runoff (Tr): it is also obtained from the distributed model, and the average value is represented in the hyetograph of Figure 7 as the sum of two terms: runoff from rainfall and runoff from snowmelt.
Table 4.Benchmarking of the water budget (in the range 0-200 hours) in the stream gauges of Isaba and Sigues.
|
| Isaba | Sigues | Isaba | Sigues | Isaba / Sigues |
|
| hm3 | hm3 | mm | mm | % |
Tp | Total precipitation | 31.79 | 67.61 | 168 | 134 | 127% |
Tr | Total runoff | 17.90 | 42.06 | 95 | 83 | 90% |
Ep | - Effective precipitation | 11.47 | 19.27 | 61 | 38 | 101% |
Sm | - Snowmelt | 6.43 | 22.79 | 34 | 45 | 76% |
L | Losses | 20.32 | 48.33 | 107 | 95 | 148% |
Hv* | Hydrograph volume | 9.80 | 25.86 | 52 | 51 | 80% |
(*)It is the volume measured between 0 and 200 hours. On the other hand, it remains to leave the basin (in Sigues stream gauge) 42.06-25.86 = 16.2 adding the volume prior to the start of the event.
In short, the terms Tp, Tr and Sm are obtained from data and direct results of the model. The terms Ep and L can be obtained from equations 11 and 12, which are partial water balance equations:
(1) Effective precipitation (Ep): in Figure 7 it comes as runoff from rain.
Ep = Tr – Sm (11)
(2) Losses (L): It is the difference between the total precipitation and the effective precipitation.
L = Tp – Ep (12)
The volume of the hydrograph (Hv) is obtained by integration of the actual hydrograph, in the 0-200 h range. The volume of water that remains in the basin just at the 200 h coordinate is the difference between total runoff (Tr) and volume of the hydrograph (Hv) adding the volume prior to the start of the episode. From the comparison of Table 4, analysing the specific values (mm), the following is deduced:
(1) Based on observed data: in Isaba there is a greater proportion of gross precipitation, although the output is similar (consider the volume of output = volume of the hydrographs).
(2) As data from the simulation: in Isaba (with a higher average altitude of the basin), there is less thaw and greater losses, part of which can be attributed to snow retention.”
Figures 1 and 3 are preserved in colour and both figures can be visualized in grey scale. The other figures have been passed to different shades of grey scale to facilitate the reader to properly identify the content.
We have made an important structural and background reform in the introduction, adding more than 40 new bibliographical references. Below are the new references that, in our opinion, are of international scope and cover several topics related to the snowmelt:
Duan, Y.C.; Liu, T.; Meng, F.H.; Luo, M: Frankl, A.; De Maeyer, Ph.; Bao, A.; Kurban, A.; Feng, X. W. Inclusion of Modified Snow Melting and Flood Processes in the SWAT Model. Water 2018, 10, 1715; doi:10.3390/w10121715
Steimke, A. L.; Han, B.S.; Brandt J. S.; Flores A.N. Climate Change and Curtailment: Evaluating Water Management Practices in the Context of Changing Runoff Regimes in a Snowmelt-Dominated Basin. Water 2018, 10, 1490; doi:10.3390/w10101490
Shen, Y. J.; Shen, Y; Fink, M.; Kralisch, S.; Chen, Y.; Brenning, A. Trends and variability in streamflow and snowmelt runoff timing in the southern Tianshan Mountains. J. Hydrol. 2018. Volume 557, Pages 173-181. doi.org/10.1016/j.jhydrol.2017.12.035
Dudley, R.W. ; Hodgkins, G.A. ; McHale, M.R.; Kolian, M.J.; Renard, M.J. Trends in snowmelt-related streamflow timing in the conterminous United States.J. Hydrol. 2017. Volume 547, Pages 208-221. doi.org/10.1016/j.jhydrol.2017.01.051
Penna, D. ; van Meerveld, H.J.; Zuecco, G.; Dalla Fontana, G.; Borga, M. Hydrological response of an Alpine catchment to rainfall and snowmelt events. J. Hydrol. 2016. Volume 537, Pages 382-397. doi.org/10.1016/j.jhydrol.2016.03.040
Vormoor, K.; Lawrence, D.; Schlichting, L.; Wilson, D.; Wong, W. K. Evidence for changes in the magnitude and frequency of observed rainfall vs. snowmelt driven floods in Norway. J. Hydrol.2016. Volume 538, Pages 33-48. doi.org/10.1016/j.jhydrol.2016.03.066
Kult, J.; Choi, W.; Choi, J. Sensitivity of the Snowmelt Runoff Model to snow covered area and temperature inputs. Appl. Geogr. 2014, 55, 30-38
Driscoll, J. M.; Meixner, T.; Molotch, N. P.; Ferre, T. P. A.; Williams, M. W.; Sickman, J. O. Event-Response Ellipses: A Method to Quantify and Compare the Role of Dynamic Storage at the Catchment Scale in Snowmelt-Dominated Systems. Water 2018, 10, 1824; doi:10.3390/w10121824
Duan L. L. and Cai, T. J. Changes in Magnitude and Timing of High Flows in Large Rain-DominatedWatersheds in the Cold Region of North-Eastern China. Water 2018, 10, 1658; doi:10.3390/w10111658
Littell, J.S.; McAfee, A. A.; Hayward G. D. Alaska Snowpack Response to Climate Change: Statewide Snowfall Equivalent and Snowpack Water Scenarios. Water 2018, 10, 668; doi:10.3390/w10050668
Kienzle, S.W. A new temperature based method to separate rain and snow. Hydrol. Process. 2008, 22, 5067–5085.
Legates, D.R.; Bogart, T.A.; Legates, D.R.; Bogart, T.A. Estimating the Proportion of Monthly Precipitation that Falls in Solid Form. J. Hydrometeorol. 2009, 10, 1299–1306.
Pistocchi, A.; Bagli, A.; Callegari, M.; Notarnicola, C.; Mazzoli, P. On the Direct Calculation of Snow Water Balances Using Snow Cover Information. Water 2017, 9, 848; doi:10.3390/w9110848
DeBeer, C. M. and Pomeroy, J. W. Influence of snowpack and melt energy heterogeneity on snow cover depletion and snowmelt runoff simulation in a cold mountain environment. J. Hydrol. 2017. Volume 553, Pages 199-213. doi.org/10.1016/j.jhydrol.2017.07.051
Xie, S. P.; Du, J.; Zhou, X. B.; Zhang, X. L.; Feng, X. Z.; Zheng, W. L.; Li, Z. G.; Xu C. Y. A progressive segmented optimization algorithm for calibrating time-variant parameters of the snowmelt runoff model (SRM). J. Hydrol. 2018. Volume 566, Pages 470-483. doi.org/10.1016/j.jhydrol.2018.09.030
Flynn, K.F. Evaluation of Swat for Sediment Prediction in a Mountainous Snowmelt-Dominated Catchment. Trans. Asabe 2011, 54, 113–122.
Kim, S.B.; Shin, H.J.; Park, M.; Kim, S.J. Assessment of Future Climate Change Impacts on Snowmelt and Stream Water Quality for a Mountainous High-Elevation Watershed Using Swat. Paddy Water Environ. 2015, 13, 557–569.
Zhang, X.Y.; Jia, L.I.; Yang, Y.Z.; You, Z. Runoff Simulation of the Catchment of the Headwaters of the Yangtze River Based on Swat Model. J. Northwest For. Univ. 2012, 5, 009.
Yu,W.; Zhao, Y.; Nan, Z.; Li, S. Improvement of Snowmelt Implementation in the Swat Hydrologic Model. Acta Ecol. Sin. 2013, 33, 6992–7001.
Arnold, J.G.; Fohrer, N. Swat2000: Current Capabilities and Research Opportunities in AppliedWatershed Modelling. Hydrol. Process. 2005, 19, 563–572.
Fontaine, T.A.; Cruickshank, T.S.; Arnold, J.G.; Hotchkiss, R.H. Development of a Snowfall–Snowmelt Routine for Mountainous Terrain for the SoilWater Assessment Tool (Swat). J. Hydrol. 2002, 262, 209–223.
Wang, X.; Melesse, A.M. Evaluation of the Swat Model’s Snowmelt Hydrology in a Northwestern Minnesota Watershed. Trans. Asabe 2005, 48, 1359–1376.
Meng, X.; Ji, X.; Liu, Z.; Xiao, J.; Chen, X.; Wang, F. Research on Improvement and Application of Snowmelt Module in Swat. J. Nat. Resour. 2014, 29, 528–539.
Fuka, D.R.; Easton, Z.M.; Brooks, E.S.; Boll, J.; Steenhuis, T.S.;Walter, M.T. A Simple Process-Based Snowmelt Routine to Model Spatially Distributed Snow Depth and Snowmelt in the Swat Model. Jawra J. Am. Water Resour. Assoc. 2012, 48, 1151–1161.
Green, C.H.; Van Griensven, A. Autocalibration in Hydrologic Modeling: Using Swat2005 in Small-Scale Watersheds. Environ. Model. Softw. 2008, 23, 422–434.
Ahl, R.S.; Woods, S.W.; Zuuring, H.R. Hydrologic Calibration and Validation of Swat in a Snow-Dominated Rocky Mountain Watershed, Montana, U.S.A. Jawra J. Am. Water Resour. Assoc.2010, 44, 1411–1430.
Haq, M. Snowmelt Runoff Investigation in River Swat Upper Basin Using Snowmelt Runoff Model, Remote Sensing and GIS Techniques; The International Institute for Geo-information Science and Earth: Enschede, The Netherlands, 2008.
Ficklin, D.L.; Barnhart, B.L. Corrigendum to “Swat Hydrologic Model Parameter Uncertainty and Its Implications for Hydroclimatic Projections in Snowmelt-Dependent Watersheds. J. Hydrol.2015, 527, 1189.
Dahri, Z.H.; Ahmad, B.; Leach, J.H.; Ahmad, S. Satellite-Based Snowcover Distribution and Associated Snowmelt Runoff Modeling in Swat River Basin of Pakistan.Proc. Pak. Acad. Sci. 2011, 48, 19–32.
Zhou, Z.; Bi, Y. Improvement of Swat Model and Its Application in Simulation of Snowmelt Runoff. In Proceedings of the National Symposium on Ice Engineering, Hohhot, China, 1 July 2011.
Sexton, A.M.; Sadeghi, A.M.; Zhang, X.; Srinivasan, R.; Shirmohammadi, A. Using Nexrad and Rain Gauge Precipitation Data for Hydrologic Calibration of Swat in a Northeastern Watershed. Trans. ASABE 2010, 53, 1501–1510.
Corripio, J. G. and López-Moreno, J. I. Analysis and Predictability of the Hydrological Response of Mountain Catchments to Heavy Rain on Snow Events: A Case Study in the Spanish Pyrenees. Hydrology2017, 4, 20; doi:10.3390/hydrology4020020
Berezowski, T. and Chybicki, A. High-Resolution Discharge Forecasting for Snowmelt and Rainfall Mixed Events. Water 2018, 10, 56; doi:10.3390/w10010056
Wu, Y. Y.; Ouyang, W.; Hao, Z. C.; Yang, B.; Wang, L. Snowmelt water drives higher soil erosion than rainfall water in a mid-high latitude upland watershed. J. Hydrol. 2018. Volume 556, Pages 438-448. doi.org/10.1016/j.jhydrol.2017.11.037
Starkloff, T.; Hessel, R.; Stolte, J.; Ritsema, C. Catchment Hydrology during Winter and Spring and the Link to Soil Erosion: A Case Study in Norway. Hydrology 2017, 4, 15; doi:10.3390/hydrology4010015
Hansen, N.C.; Gupta, S.C.; Moncrief, J.F. Snowmelt runoff, sediment, and phosphorous losses under three different tillage systems. Soil Tillage Res. 2000, 57, 93–100.
Govers, G. Rill erosion on arable land in central Belgium: Rates, controls and predictability. Catena 1991, 18, 133–155.
Boardman, J.; Shepheard, M.L.;Walker, E.; Foster, I.D.L. Soil erosion risk-assessment for on- and off-farm impacts: A test case using the Midhurst area, West Sussex, UK. J. Environ. Manag.2009, 90, 2578–2588.
Weigert, A.; Schmidt, J.Water transport under winter conditions. Catena 2005, 64, 193–208.
Yakutina, O.P.; Nechaeva, T.V.; Smirnova, N.V. Consequences of snowmelt erosion: Soil fertility, productivity and quality of wheat on Greyzemic Phaeozem in the south of West Siberia. Agric. Ecosyst. Environ. 2015, 200, 88–93.
Su, J.J.; van Bochove, E.; Thériault, G.; Novotma, B.; Khaldoune, J.; Denault, J.T.; Zhou, J.; Nolin, M.C.; Hu, C.X.; Bernier, M.; et al. Effects of snowmelt on phosphorus and sediment losses from agricultural watersheds in Eastern Canada. Agric. Water Manag. 2011, 98, 867–876.
Rivera, J. A.; Penalba, O. C; Villalba, R.; Araneo, D. C. Spatio-Temporal Patterns of the 2010–2015 Extreme Hydrological Drought across the Central Andes, Argentina. Water 2017, 9, 652; doi:10.3390/w9090652
Stock, M. N.; Arriaga, F. J.; Vadas, P. A.; Karthikeyan, K. G. Manure application timing drives energy absorption for snowmelt on an agricultural soil. J. Hydrol. 2019, 569, 51-60. doi.org/10.1016/j.jhydrol.2018.11.028
We deeply appreciate the time spent by the Reviewer, who will undoubtedly contribute to the improvement of our manuscript.

Reviewer 2 Report
Comments on “Assessment of snowmelt role for a flood event in a gauged catchment” by Mateo-Lazaro et al.
The paper presents the results of a theoretical application aimed at understanding the role of snowmelt in the interpretation of a flood event. The study has been carried out considering a well-monitored test case.
The findings are attractive even if they are presented unconvincingly. The English is of poor quality. The main problems are:
1) Lack of a consolidated international literature
2) Poor quality of images that if printed in black and white are not clear
3) The introduction is a labyrinth of information that does not follow a common thread
Minor comments
Line 35-37: not use passive form (remember not article but paper)
Line 40: there is a repetition of a sentence
Line 67: reference for SHEE
Line 100: Where and not were
Line 138: check symbol of equation 7
Line 166: the experiment arrangement is not clear (number of gauges and pluviometer is confused)
Table 2 is not clear
Line 182: lack of references for RBF
Figure 5 not clear without coulors
I suggest merging figure 7 and 8 to compare Temperature evolution with runoff from snowmelt
Paragraph 4.3 has to be written completely as it is not clear (methodology, units of measures and which kind of data you used)
Figure 11: legend in English
I suggest a deep revision and in some case a rewritten of many part to make the results fascinating
Author Response
Dear Reviewers and Editors,
We would like to thank three anonymous Reviewers for having handled and reviewed our manuscript. We really appreciate your comments to improve this manuscript. We would like to show you what we have revised following your comments. In this letter, in blue you can read our reply to your kind comments.
In the file “water-427845_marked_manuscript”, the corrections are in red.
In the file "water-427845_Clean_manuscript_Sami revised" in blue, the changes of the language revision are marked.
The final version with all the changes is in the file "water-427845_Clean_manuscript".
Reviewer 2:
Comments on “Assessment of snowmelt role for a flood event in a gauged catchment” by Mateo-Lazaro et al.
The paper presents the results of a theoretical application aimed at understanding the role of snowmelt in the interpretation of a flood event. The study has been carried out considering a well-monitored test case.
The findings are attractive even if they are presented unconvincingly. The English is of poor quality. The main problems are:
Response
Thank you for your comments. We have tried to improve English throughout the paper. In the marked manuscript, we have with red colour for substantial changes of content.
In the file “water-427845_Clean_manuscript_Sami revised” in blue colour are the language corrections.
1) Lack of a consolidated international literature
Response:
The authors appreciate the reviewer’s observation. We have made an important structural and background reform in the introduction, adding more than 40 new bibliographical references. Below are the new references that, in our opinion, are of international scope and cover several topics related to the snowmelt:
Duan, Y.C.; Liu, T.; Meng, F.H.; Luo, M: Frankl, A.; De Maeyer, Ph.; Bao, A.; Kurban, A.; Feng, X. W. Inclusion of Modified Snow Melting and Flood Processes in the SWAT Model. Water 2018, 10, 1715; doi:10.3390/w10121715
Steimke, A. L.; Han, B.S.; Brandt J. S.; Flores A.N. Climate Change and Curtailment: Evaluating Water Management Practices in the Context of Changing Runoff Regimes in a Snowmelt-Dominated Basin. Water 2018, 10, 1490; doi:10.3390/w10101490
Shen, Y. J.; Shen, Y; Fink, M.; Kralisch, S.; Chen, Y.; Brenning, A. Trends and variability in streamflow and snowmelt runoff timing in the southern Tianshan Mountains. J. Hydrol. 2018. Volume 557, Pages 173-181. doi.org/10.1016/j.jhydrol.2017.12.035
Dudley, R.W. ; Hodgkins, G.A. ; McHale, M.R.; Kolian, M.J.; Renard, M.J. Trends in snowmelt-related streamflow timing in the conterminous United States.J. Hydrol. 2017. Volume 547, Pages 208-221. doi.org/10.1016/j.jhydrol.2017.01.051
Penna, D. ; van Meerveld, H.J.; Zuecco, G.; Dalla Fontana, G.; Borga, M. Hydrological response of an Alpine catchment to rainfall and snowmelt events. J. Hydrol. 2016. Volume 537, Pages 382-397. doi.org/10.1016/j.jhydrol.2016.03.040
Vormoor, K.; Lawrence, D.; Schlichting, L.; Wilson, D.; Wong, W. K. Evidence for changes in the magnitude and frequency of observed rainfall vs. snowmelt driven floods in Norway. J. Hydrol. 2016. Volume 538, Pages 33-48. doi.org/10.1016/j.jhydrol.2016.03.066
Kult, J.; Choi, W.; Choi, J. Sensitivity of the Snowmelt Runoff Model to snow covered area and temperature inputs. Appl. Geogr. 2014, 55, 30-38
Driscoll, J. M.; Meixner, T.; Molotch, N. P.; Ferre, T. P. A.; Williams, M. W.; Sickman, J. O. Event-Response Ellipses: A Method to Quantify and Compare the Role of Dynamic Storage at the Catchment Scale in Snowmelt-Dominated Systems. Water 2018, 10, 1824; doi:10.3390/w10121824
Duan L. L. and Cai, T. J. Changes in Magnitude and Timing of High Flows in Large Rain-DominatedWatersheds in the Cold Region of North-Eastern China. Water 2018, 10, 1658; doi:10.3390/w10111658
Littell, J.S.; McAfee, A. A.; Hayward G. D. Alaska Snowpack Response to Climate Change: Statewide Snowfall Equivalent and Snowpack Water Scenarios. Water 2018, 10, 668; doi:10.3390/w10050668
Kienzle, S.W. A new temperature based method to separate rain and snow. Hydrol. Process. 2008, 22, 5067–5085.
Legates, D.R.; Bogart, T.A.; Legates, D.R.; Bogart, T.A. Estimating the Proportion of Monthly Precipitation that Falls in Solid Form. J. Hydrometeorol. 2009, 10, 1299–1306.
Pistocchi, A.; Bagli, A.; Callegari, M.; Notarnicola, C.; Mazzoli, P. On the Direct Calculation of Snow Water Balances Using Snow Cover Information. Water 2017, 9, 848; doi:10.3390/w9110848
DeBeer, C. M. and Pomeroy, J. W. Influence of snowpack and melt energy heterogeneity on snow cover depletion and snowmelt runoff simulation in a cold mountain environment. J. Hydrol. 2017. Volume 553, Pages 199-213. doi.org/10.1016/j.jhydrol.2017.07.051
Xie, S. P.; Du, J.; Zhou, X. B.; Zhang, X. L.; Feng, X. Z.; Zheng, W. L.; Li, Z. G.; Xu C. Y. A progressive segmented optimization algorithm for calibrating time-variant parameters of the snowmelt runoff model (SRM). J. Hydrol. 2018. Volume 566, Pages 470-483. doi.org/10.1016/j.jhydrol.2018.09.030
Flynn, K.F. Evaluation of Swat for Sediment Prediction in a Mountainous Snowmelt-Dominated Catchment. Trans. Asabe 2011, 54, 113–122.
Kim, S.B.; Shin, H.J.; Park, M.; Kim, S.J. Assessment of Future Climate Change Impacts on Snowmelt and Stream Water Quality for a Mountainous High-Elevation Watershed Using Swat. Paddy Water Environ. 2015, 13, 557–569.
Zhang, X.Y.; Jia, L.I.; Yang, Y.Z.; You, Z. Runoff Simulation of the Catchment of the Headwaters of the Yangtze River Based on Swat Model. J. Northwest For. Univ. 2012, 5, 009.
Yu,W.; Zhao, Y.; Nan, Z.; Li, S. Improvement of Snowmelt Implementation in the Swat Hydrologic Model. Acta Ecol. Sin. 2013, 33, 6992–7001.
Arnold, J.G.; Fohrer, N. Swat2000: Current Capabilities and Research Opportunities in AppliedWatershed Modelling. Hydrol. Process. 2005, 19, 563–572.
Fontaine, T.A.; Cruickshank, T.S.; Arnold, J.G.; Hotchkiss, R.H. Development of a Snowfall–Snowmelt Routine for Mountainous Terrain for the SoilWater Assessment Tool (Swat). J. Hydrol. 2002, 262, 209–223.
Wang, X.; Melesse, A.M. Evaluation of the Swat Model’s Snowmelt Hydrology in a Northwestern Minnesota Watershed. Trans. Asabe 2005, 48, 1359–1376.
Meng, X.; Ji, X.; Liu, Z.; Xiao, J.; Chen, X.; Wang, F. Research on Improvement and Application of Snowmelt Module in Swat. J. Nat. Resour. 2014, 29, 528–539.
Fuka, D.R.; Easton, Z.M.; Brooks, E.S.; Boll, J.; Steenhuis, T.S.;Walter, M.T. A Simple Process-Based Snowmelt Routine to Model Spatially Distributed Snow Depth and Snowmelt in the Swat Model. Jawra J. Am. Water Resour. Assoc. 2012, 48, 1151–1161.
Green, C.H.; Van Griensven, A. Autocalibration in Hydrologic Modeling: Using Swat2005 in Small-Scale Watersheds. Environ. Model. Softw. 2008, 23, 422–434.
Ahl, R.S.; Woods, S.W.; Zuuring, H.R. Hydrologic Calibration and Validation of Swat in a Snow-Dominated Rocky Mountain Watershed, Montana, U.S.A. Jawra J. Am. Water Resour. Assoc.2010, 44, 1411–1430.
Haq, M. Snowmelt Runoff Investigation in River Swat Upper Basin Using Snowmelt Runoff Model, Remote Sensing and GIS Techniques; The International Institute for Geo-information Science and Earth: Enschede, The Netherlands, 2008.
Ficklin, D.L.; Barnhart, B.L. Corrigendum to “Swat Hydrologic Model Parameter Uncertainty and Its Implications for Hydroclimatic Projections in Snowmelt-Dependent Watersheds. J. Hydrol.2015, 527, 1189.
Dahri, Z.H.; Ahmad, B.; Leach, J.H.; Ahmad, S. Satellite-Based Snowcover Distribution and Associated Snowmelt Runoff Modeling in Swat River Basin of Pakistan. Proc. Pak. Acad. Sci. 2011, 48, 19–32.
Zhou, Z.; Bi, Y. Improvement of Swat Model and Its Application in Simulation of Snowmelt Runoff. In Proceedings of the National Symposium on Ice Engineering, Hohhot, China, 1 July 2011.
Sexton, A.M.; Sadeghi, A.M.; Zhang, X.; Srinivasan, R.; Shirmohammadi, A. Using Nexrad and Rain Gauge Precipitation Data for Hydrologic Calibration of Swat in a Northeastern Watershed. Trans. ASABE 2010, 53, 1501–1510.
Corripio, J. G. and López-Moreno, J. I. Analysis and Predictability of the Hydrological Response of Mountain Catchments to Heavy Rain on Snow Events: A Case Study in the Spanish Pyrenees. Hydrology2017, 4, 20; doi:10.3390/hydrology4020020
Berezowski, T. and Chybicki, A. High-Resolution Discharge Forecasting for Snowmelt and Rainfall Mixed Events. Water 2018, 10, 56; doi:10.3390/w10010056
Wu, Y. Y.; Ouyang, W.; Hao, Z. C.; Yang, B.; Wang, L. Snowmelt water drives higher soil erosion than rainfall water in a mid-high latitude upland watershed. J. Hydrol. 2018. Volume 556, Pages 438-448. doi.org/10.1016/j.jhydrol.2017.11.037
Starkloff, T.; Hessel, R.; Stolte, J.; Ritsema, C. Catchment Hydrology during Winter and Spring and the Link to Soil Erosion: A Case Study in Norway. Hydrology 2017, 4, 15; doi:10.3390/hydrology4010015
Hansen, N.C.; Gupta, S.C.; Moncrief, J.F. Snowmelt runoff, sediment, and phosphorous losses under three different tillage systems. Soil Tillage Res. 2000, 57, 93–100.
Govers, G. Rill erosion on arable land in central Belgium: Rates, controls and predictability. Catena 1991, 18, 133–155.
Boardman, J.; Shepheard, M.L.;Walker, E.; Foster, I.D.L. Soil erosion risk-assessment for on- and off-farm impacts: A test case using the Midhurst area, West Sussex, UK. J. Environ. Manag.2009, 90, 2578–2588.
Weigert, A.; Schmidt, J.Water transport under winter conditions. Catena 2005, 64, 193–208.
Yakutina, O.P.; Nechaeva, T.V.; Smirnova, N.V. Consequences of snowmelt erosion: Soil fertility, productivity and quality of wheat on Greyzemic Phaeozem in the south of West Siberia. Agric. Ecosyst. Environ. 2015, 200, 88–93.
Su, J.J.; van Bochove, E.; Thériault, G.; Novotma, B.; Khaldoune, J.; Denault, J.T.; Zhou, J.; Nolin, M.C.; Hu, C.X.; Bernier, M.; et al. Effects of snowmelt on phosphorus and sediment losses from agricultural watersheds in Eastern Canada. Agric. Water Manag. 2011, 98, 867–876.
Rivera, J. A.; Penalba, O. C; Villalba, R.; Araneo, D. C. Spatio-Temporal Patterns of the 2010–2015 Extreme Hydrological Drought across the Central Andes, Argentina. Water 2017, 9, 652; doi:10.3390/w9090652
Stock, M. N.; Arriaga, F. J.; Vadas, P. A.; Karthikeyan, K. G. Manure application timing drives energy absorption for snowmelt on an agricultural soil. J. Hydrol. 2019, 569, 51-60. doi.org/10.1016/j.jhydrol.2018.11.028
2) Poor quality of images that if printed in black and white are not clear
Response:
The authors appreciate the reviewer’s observation. Figures 1 and 3 are preserved in colour. Both figures can be visualized in grey scale as shown below. The other figures have been passed to different shades of grey scale to facilitate the reader to properly identify the content.
Figure 1 in grey scale:
Figure 3 in grey scale:
3) The introduction is a labyrinth of information that does not follow a common thread
Response:
Thank you for the comments. We have made an important structural and background reform in the introduction. The introductory framework and its references are carried out sequentially ordered by theme. The changes are the following:
“Hydrological processes related to snow (snowfall, snowpack, snowmelt) occur in large regions of the Earth. The snowmelt is one of the processes that intervene in the hydrological cycle interacting with many other processes. This article focuses on the phenomenon of melt-induced flooding and the changes it causes in the runoff regime, but this issue and its consequences has already been widely studied from a multitude of approaches [1, 2].
The tendency and the variability of the discharge in the rivers and their relation with the runoff coming from the snowmelt is a recurrent topic even in recent studies [3-6]. Sensitivity studies of the variables that intervene in the melting processes is another key issue, for example Reference [7] evaluates the sensitivity of a snowmelt runoff model with temperature input data, in a region with complex temperature elevation gradients.
Another phenomenon to be highlighted caused by the melting process is the time lag in the water balance, a lag that takes place between precipitation in the form of snow and runoff, which is another of the topics we quantify in this article. Recently other authors have contributed light on this topic [8, 9]. Other authors have developed rain-snow separation methods within time series and their lag with runoff [10-12]. Reference [13] uses data from the temporary estimate of the thickness of the snow cover to study the time lag. Reference [14] models the relationships of the depletion of the snow cover with the influence of the heterogeneity of the accumulation of snow and with the melting energy. Reference [15] proposes calibration methods for the time lag snow-runoff.
In addition to the methods proposed in [23] there are recent publications that use models developed to implement the snow-runoff process. The well-known SWAT model is widely used also in recent publications [1, 16-31]. Reference [15] develops a progressive segmented optimization algorithm to calibrate the temporal variation parameters of the snowfall-runoff model. Reference [14] conducts simulations of snowfall-runoff in various environments of mountain basins. The most frequent parameters used in the models are analysed by reference [32] which concludes with the development of hydrological basin models with runoff from the thaw and shows that substantial progress has been made with models of different sophistication.
Other authors focus their research on the forecast of melting processes and their consequences. Reference [33] performs prediction studies with thaw in the Spanish Pyrenean chain, research location area of our article. In other cases, the studies focus on urbanized areas such as Reference [34] where the effect of snow melting on discharge forecasts is highlighted, carrying out a study in an urbanized basin with clear influence of snow processes on the increase of runoff.
Above we have referred to the relationship of the melting process with other processes. Erosion and sediment transport are processes that maintain a relationship. In many cases water melted by snow causes greater erosion of the soil than rainwater [35] so it is appropriate to investigate the differences in soil infiltration capacity during periods of rain or melting. Soil erosion in agricultural areas during winter and spring is a problem in many countries of the world such as Norway [36], USA [37], Belgium [38], United Kingdom [39], Germany [40] and Russia [41]. In these areas, soil erosion during winter and spring depletes the nutrient-rich top layer and contributes elements (phosphorus, nitrogen) to freshwater bodies [42].
The period of droughts is another process that may be related to the melting processes, in this case due to an absence or prolonged decrease in the contribution of snow [43].
Energy and thaw are also processes related to each other. Reference [44] shows how changes in the composition or structure of the soil in large areas influence the energy input of the soil and therefore the rate of thawing. Reference [14] demonstrates the influence of spatial distribution of snow cover thickness and heterogeneity in the distribution of fusion energy in extreme runoff production.
It can be said that the snow and thaw processes play an important role in the hydrological cycle, both in themselves and by the direct relationship with other processes, as well as by their regulatory role of water flow and volume of water reserve.”
Minor comments
Line 35-37: not use passive form (remember not article but paper)
Response: Thanks for the comment. We have changed the phrase as follows:
“The process of snowmelt has been considered in this paper within a flood model with hydro-meteorological methods or models of rainfall-runoff. We have introduced the melting process as an additional contribution that flows through the hydrological system of the basin.”
Line 40: there is a repetition of a sentence
Response: Thanks for the comment. This part of the text has been deleted when rewriting the Introduction section.
Line 67: reference for SHEE
Response: We have provided the link "http://www.unizar.es/hidrologia" where you can download an example. On the other hand, reference to papers is indicated below [45, 46, 53-61] where software parts are described. A more exhaustive description has also been added within the Methods section:
“The software SHEE has numerous applications for either DEM management or hydrological processes simulation [62]. Obtaining new cartographic coverage with the combination of DEM and simulated processes is also possible. The DEM management is achieved using the GDAL (Geospatial Data Abstraction Library), which permits to import and export different archive formats and to make new coverage from multiple archives. The program can combine coverage with different coordinate system thanks to the use of the PROJ4 library from the USGS. Thousands of terrestrial geodetic systems can be represented, transformed and converted between them. To do that, the program is able to obtain necessary Spatial Reference Organization parameters from the internet server transfer. Downloading information from WMS remote server is also possible. With regard to DEM characteristics, SHEE program can manage any format, size, accuracy and reference system. E.g., Global DEM has been used like SRTM30 (with file size 3.6 GB and grid size 30”, ≈900 m), MDT5 of Spanish territory (120 GB and 5 m) and Lidar. The use of PLR models (Parallel Linear Reservoir) as a hydrological model integrated within the sequential processing algorithm of the catchment is a special case of hydrological application where every cell of the DEM is considered as a reservoir combination in parallel [63].
Future developments of SHEE program are very promising. Recently we have incorporated a module for development and visualization of geological structures in three dimensions. In the near future, it will allow the development of combined hydrogeological models with complex subsurface structures. This implementation has resulted in the publication Reference [55]. Therein, we present an application that visualizes three-dimensional geological structures with digital terrain models. The three-dimensional structures are displayed as their intersection with two-dimensional surfaces that may be defined analytically (e.g., sections) or with grid meshes in the case of irregular surfaces such as the digital terrain models. Additionally, the process of generating new textures can be performed by a Graphics Processing Unit (GPU), thereby making real-time processing very effective and providing the possibility of displaying the simulation of geological structures in motion. Regarding the Graphics Processing Units, and since the DEM is becoming denser, we are currently completing the development of hydrological models with this technique through the sequential processing algorithm, whose main advantage is the shortening of the computation time, which can be reduced 100 times. Due to parallel processing, it is necessary to reprogram the sequential algorithms for computing drainage networks.”
Line 100: Where and not were
Response: Thank you, sorry for the error.
Line 138: check symbol of equation 7
Response: Thank you it’s about alpha (a).
Line 166: the experiment arrangement is not clear (number of gauges and pluviometer is confused)
Response: Thanks for the comment. We have tried to make it clearer with the following sentence:
“Altimetry representation of the Esca river basin with the rain gauges (A063, A268, A259 and P016) and the two stream gauges (A063, A268) that also have rain gauges.“
Table 2 is not clear
Response: Thanks for the observation, we have changed table 2 as follows:
Date |
Rain duration |
Time interval |
Peak flow | |
Start time |
End time |
hours |
15 minutes |
m3/s |
January 18, 2009, 2:30 p.m. |
January 28, 2009, 7:30 p.m. |
245.25 |
981 |
201 |
Line 182: lack of references for RBF
Response: Thanks for the observation. We have added the following references of recent publication:
Tayyab, M.; Ahmad, I.; Sun, N.; Zhou, J.; Dong, X. Application of Integrated Artificial Neural Networks Based on Decomposition Methods to Predict Streamflow at Upper Indus Basin, Pakistan. Atmosphere.2018, 9, 494; doi:10.3390/atmos9120494
Koycegiz, C. and Buyukyildiz, M. Calibration of SWAT and Two Data-Driven Models for a Data-Scarce Mountainous Headwater in Semi-Arid Konya Closed Basin.Water. 2019, 11, 147; doi:10.3390/w11010147
Ma, M.; Liu, C.; Zhao, G.; Xie, H.; Jia, P.; Wang, D.; Wang, H.; Hong, Y. Flash Flood Risk Analysis Based on Machine Learning Techniques in the Yunnan Province, China. Remote Sens. 2019, 11, 170; doi:10.3390/rs11020170
Figure 5 not clear without coulors
Response: Thanks for the observation. Figure 5 has been converted to grey scale:
I suggest merging figure 7 and 8 to compare Temperature evolution with runoff from snowmelt
Response: Thanks for the comment. We have put these two figures together in one:
Paragraph 4.3 has to be written completely as it is not clear (methodology, units of measures and which kind of data you used)
Response: Thank you for the comments. Section 4.3 has been rewritten, taking special care in the units and in the origin of the data used in the water balance:
“Table 4 contains the results of the water balance of the episode studied considering the temporal section between 0 and 200 hours. The balance is carried out in the two gauging stations (Isaba and Sigues), and the results are given in total units (hm3) for the entire basin corresponding to each gauging station. They are also given in specific units (mm), that is, total units dividing by the area of the corresponding basin. The last column represents a comparison between the results of the two gauging stations.
To elaborate the water balance, we start from the data and results of hyetographs and hydrographs that we have been using in modelling and that are summarized in three components:
(1) Total precipitation (Tp): It has been obtained from the model of distributed precipitation based on functions (RBF) shown in Figure 4, adding all the time intervals every 15 minutes from 0 to 200 hours. The results of this integration coincide with the term Total Rainfall of the average hyetograph of Figures 5 and 7.
(2) Snowmelt (Sm): it is determined in the distributed model calculated with the SHEE program, whose average representation coincides with the snowmelt fraction of the average hyetograph of Figure 7.
(3) Total runoff (Tr): it is also obtained from the distributed model, and the average value is represented in the hyetograph of Figure 7 as the sum of two terms: runoff from rainfall and runoff from snowmelt.
Table 4.Benchmarking of the water budget (in the range 0-200 hours) in the stream gauges of Isaba and Sigues.
|
|
Isaba |
Sigues |
Isaba |
Sigues |
Isaba / Sigues |
|
|
hm3 |
hm3 |
mm |
mm |
% |
Tp |
Total precipitation |
31.79 |
67.61 |
168 |
134 |
127% |
Tr |
Total runoff |
17.90 |
42.06 |
95 |
83 |
90% |
Ep |
- Effective precipitation |
11.47 |
19.27 |
61 |
38 |
101% |
Sm |
- Snowmelt |
6.43 |
22.79 |
34 |
45 |
76% |
L |
Losses |
20.32 |
48.33 |
107 |
95 |
148% |
Hv* |
Hydrograph volume |
9.80 |
25.86 |
52 |
51 |
80% |
(*)It is the volume measured between 0 and 200 hours. On the other hand, it remains to leave the basin (in Sigues stream gauge) 42.06-25.86 = 16.2 adding the volume prior to the start of the event.
In short, the terms Tp, Tr and Sm are obtained from data and direct results of the model. The terms Ep and L can be obtained from equations 11 and 12, which are partial water balance equations:
(1) Effective precipitation (Ep): in Figure 7 it comes as runoff from rain.
Ep = Tr – Sm (11)
(2) Losses (L): It is the difference between the total precipitation and the effective precipitation.
L = Tp – Ep (12)
The volume of the hydrograph (Hv) is obtained by integration of the actual hydrograph, in the 0-200 h range. The volume of water that remains in the basin just at the 200 h coordinate is the difference between total runoff (Tr) and volume of the hydrograph (Hv) adding the volume prior to the start of the episode. From the comparison of Table 4, analysing the specific values (mm), the following is deduced:
(1) Based on observed data: in Isaba there is a greater proportion of gross precipitation, although the output is similar (consider the volume of output = volume of the hydrographs).
(2) As data from the simulation: in Isaba (with a higher average altitude of the basin), there is less thaw and greater losses, part of which can be attributed to snow retention.”
Figure 11: legend in English
Response: Thanks for the comment. We have modified the figure that now has the number 10.
I suggest a deep revision and in some case a rewritten of many part to make the results fascinating
Response: Thanks for the observation. We have made a substantial reform of the paper, especially in the Introduction, Methods, Data Source, Results and Discussion sections and in Bibliographic References.
We deeply appreciate the time spent by the Reviewer, who will undoubtedly contribute to the improvement of our manuscript.

Reviewer 3 Report
The manuscript titled "Assessment of snowmelt role for a flood event in a gauged catchment" deals with the very interest topic about the contribution of snowmelt in flood events, which is an open challenge and is not investigated enough.
However, it is very difficult to follow the text due to bad English. An extensive editing in English shall be done (probably from a native speaker) and reconsidered again.
The scientific quality is good but the overall picture is downgraded due to language.
Author Response
Dear Reviewers and Editors,
We would like to thank three anonymous Reviewers for having handled and reviewed our manuscript. We really appreciate your comments to improve this manuscript. We would like to show you what we have revised following your comments. In this letter, in blue you can read our reply to your kind comments.
In the file “water-427845_marked_manuscript”, the corrections are in red.
In the file "water-427845_Clean_manuscript_Sami revised" in blue, the changes of the language revision are marked.
The final version with all the changes is in the file "water-427845_Clean_manuscript".
Reviewer 3:
Comments and Suggestions for Authors
The manuscript titled "Assessment of snowmelt role for a flood event in a gauged catchment" deals with the very interest topic about the contribution of snowmelt in flood events, which is an open challenge and is not investigated enough.
However, it is very difficult to follow the text due to bad English. An extensive editing in English shall be done (probably from a native speaker) and reconsidered again.
The scientific quality is good but the overall picture is downgraded due to language.
Thanks for this suggestion.
We have made a substantial reform of the paper, especially in the Introduction, Methods, Data Source, Results and Discussion sections and in Bibliographic References.
Once the manuscript reform is finished, we have carried out an intense revision of the language whose main corrections are indicated in blue.
We deeply appreciate the time spent by the Reviewer, who will undoubtedly contribute to the improvement of our manuscript.

Round 2
Reviewer 1 Report
The paper is now in good quality, and should be interesting to readers of the Water journal.
I would suggest Authors to consider the following remarks:
1.
Row: 49-50 (in: "water-427845_Clean_manuscript") => is: Another phenomenon to be highlighted, which is also caused by the melting process, is the time lag in the water balance.
Authors may be here interested in the results of investigation conducted in small agricultural catchment [A & C] showing significantly higher values of lag time for snowmelt events than for rainfall events. In that study also relation between lag time of runoff and sediment output is presented.
2.
Row 61 => is: Reference [14] conducts simulations of snowfall-runoff;
Please reconsider the expression ‘snowfall-runoff’ with ‘snowmelt-runoff’?
3.
Row 99-100 => is: .. it is going to be our case, to use the Curve Number method of the SCS (SCS-CN model) to carry out this transformation.
The Authors may be interested to refer to earlier use of the model for snowmelt floods [B], event this model had been originally developed only for rainstorm floods.
4. References to consideration by Authors:
A. Banasik, K.; Hejduk, A. Ratio of basin lag times for runoff and sediment yield processes recorded in various environments. Sediment Dynamics From the Summit To the Sea 2014, Publ. IAHS 367, 163-169.
B. Hejduk, L.; Hejduk, A.; Banasik, K. Determination of Curve Number for snowmelt-runoff floods in a small catchment. Changes in Flood Risk and Perception in Catchments and Cities 2015, 370, 167-170, doi:10.5194/piahs-370-167-2015.
C. Hejduk, A.; Banasik, K. Recorded lag times of snowmelt events in a small catchment. Annals of Warsaw University of Life Sciences – SGGW Land Reclamation No 43 (1), 2011, 37–46, doi: 10.2478/v10060-008-0091-5

Author Response
Dear Reviewers and Editors,
We would like to thank three anonymous Reviewers for having handled and reviewed our manuscript. We really appreciate your comments to improve this manuscript. We would like to show you what we have revised following your comments. In this letter, in blue you can read our reply to your kind comments.
In the file “water-427845_marked_manuscript”, the corrections are in red.
The final version with all the changes is in the file "water-427845_Clean_manuscript"
Reviewer 1:
The paper is now in good quality, and should be interesting to readers of the Water journal.
I would suggest Authors to consider the following remarks:
1. Row: 49-50 (in: "water-427845_Clean_manuscript") => is: Another phenomenon to be highlighted, which is also caused by the melting process, is the time lag in the water balance.
Authors may be here interested in the results of investigation conducted in small agricultural catchment [A & C] showing significantly higher values of lag time for snowmelt events than for rainfall events. In that study also relation between lag time of runoff and sediment output is presented.
Response: Thanks for this suggestion. We have added the following paragraph in the introduction, and its bibliographical references:
“Reference [16, 17] investigates an agricultural small basin showing significantly higher values of lag time for snowmelt events than for rainfall events. In that study also relation between lag time of runoff and sediment output is presented.”
In References:
Banasik, K.; Hejduk, A. Ratio of basin lag times for runoff and sediment yield processes recorded in various environments. Sediment Dynamics From the Summit To the Sea. Publ. IAHS. 2014, 367, 163-169
Hejduk, A.; Banasik, K. Recorded lag times of snowmelt events in a small catchment. Annals of Warsaw University of Life Sciences – SGGW Land Reclamation No 43 (1), 2011, 37–46, doi: 10.2478/v10060-008-0091-5
2. Row 61 => is: Reference [14] conducts simulations of snowfall-runoff;
Please reconsider the expression ‘snowfall-runoff’ with ‘snowmelt-runoff’?
Response: The authors appreciate the reviewer’s observation, this expression has been changed
3. Row 99-100 => is: .. it is going to be our case, to use the Curve Number method of the SCS (SCS-CN model) to carry out this transformation.
The Authors may be interested to refer to earlier use of the model for snowmelt floods [B], event this model had been originally developed only for rainstorm floods.
Response: Thanks for this suggestion. We have added the following, and its bibliographical references:
“Other authors have taken this same decision and have used the SCS-CN model for the same purposes [53].”
In References:
Hejduk, L.; Hejduk, A.; Banasik, K. Determination of Curve Number for snowmelt-runoff floods in a small catchment. Changes in Flood Risk and Perception in Catchments and Cities, 2015, 370, 167-170, doi:10.5194/piahs-370-167-2015.
We deeply appreciate the time spent by the Reviewer, who will undoubtedly contribute to the improvement of our manuscript.

Reviewer 2 Report
Dear Authors,
thank you for your efforts in improving the manuscript.
I think that the paper could be accepted in this form after some minor check. Indeed the novelty is clear, there is a good framework with references to international works. The results are now clear. I suggest only shorting the manuscript. Indeed, I propose a reading of the text to reduce the number of words. in this way the text will be even smoother.
Regards
Author Response
Dear Reviewers and Editors,
We would like to thank three anonymous Reviewers for having handled and reviewed our manuscript. We really appreciate your comments to improve this manuscript. We would like to show you what we have revised following your comments. In this letter, in blue you can read our reply to your kind comments.
In the file “water-427845_marked_manuscript”, the corrections are in red.
The final version with all the changes is in the file "water-427845_Clean_manuscript".
Reviewer 2:
Dear Authors,
I think that the paper could be accepted in this form after some minor check. Indeed the novelty is clear, there is a good framework with references to international works. The results are now clear. I suggest only shorting the manuscript. Indeed, I propose a reading of the text to reduce the number of words. in this way the text will be even smoother.
Response: Thank you for your comments. We have carried out some reduction of the text in the Methods section
We deeply appreciate the time spent by the Reviewer, who will undoubtedly contribute to the improvement of our manuscript.
